# Graphene-Based Materials for the Separator Functionalization of Lithium-Ion/Metal/Sulfur Batteries

**DOI:** 10.3390/ma16124449

**Published:** 2023-06-18

**Authors:** Zongle Huang, Wenting Sun, Zhipeng Sun, Rui Ding, Xuebin Wang

**Affiliations:** National Laboratory of Solid State Microstructures (NLSSM), Collaborative Innovation Center of Advanced Microstructures, Jiangsu Key Laboratory of Artificial Functional Materials, College of Engineering and Applied Sciences, Nanjing University (NJU), Nanjing 210093, China; 502022340034@smail.nju.edu.cn (Z.H.); 502022340049@smail.nju.edu.cn (W.S.); 602022340041@smail.nju.edu.cn (Z.S.); dz1834014@smail.nju.edu.cn (R.D.)

**Keywords:** graphene, separator, lithium-ion battery, lithium-metal battery, lithium-sulfur battery

## Abstract

With the escalating demand for electrochemical energy storage, commercial lithium-ion and metal battery systems have been increasingly developed. As an indispensable component of batteries, the separator plays a crucial role in determining their electrochemical performance. Conventional polymer separators have been extensively investigated over the past few decades. Nevertheless, their inadequate mechanical strength, deficient thermal stability, and constrained porosity constitute serious impediments to the development of electric vehicle power batteries and the progress of energy storage devices. Advanced graphene-based materials have emerged as an adaptable solution to these challenges, owing to their exceptional electrical conductivity, large specific surface area, and outstanding mechanical properties. Incorporating advanced graphene-based materials into the separator of lithium-ion and metal batteries has been identified as an effective strategy to overcome the aforementioned issues and enhance the specific capacity, cycle stability, and safety of batteries. This review paper provides an overview of the preparation of advanced graphene-based materials and their applications in lithium-ion, lithium-metal, and lithium-sulfur batteries. It systematically elaborates on the advantages of advanced graphene-based materials as novel separator materials and outlines future research directions in this field.

## 1. Introduction

Given the escalating concerns surrounding pollution and climate change as a result of the extensive utilization of non-renewable resources, it has become crucial to prioritize the development of energy storage technologies that are sustainable, efficient, and environmentally friendly. With the decreasing cost of renewable energy sources, such as wind and solar power, rechargeable batteries have emerged as a superior option to traditional mechanical and hydraulic energy conservation due to their higher energy density and accessibility. In particular, lithium-ion batteries have gained immense popularity in various industries, including the automotive, electronic, and aerospace sectors, owing to their remarkable operating voltage, substantial energy density, minor memory effect, and eco-friendliness. Additionally, lithium-metal and -sulfur batteries, which boast higher theoretical specific capacity and lower costs, are widely regarded as one of the most promising candidates for next-generation energy storage systems.

In recent decades, the separator has developed significantly as a crucial component of lithium-ion/metal/sulfur batteries through extensive research efforts. It has been established that the separator must exhibit desirable insulation and mechanical strength to prevent short-circuiting of electrodes and the appropriate pore size and porosity to facilitate the uniform transmission of lithium ions. Furthermore, excellent thermal stability, adequate wettability, and electrolyte retention are critical to enhancing battery capacity and performance. Hence, selecting a separator with comprehensive performance is essential to improve battery performance [1,2,3,4]. Commercial battery separators typically use semi-crystalline polyolefin-based materials due to their good insulation, high corrosion resistance, low cost, and easy processing. The Celgard separator is a widely used example, consisting of a polyethylene (PS) and polypropylene (PP) composite prepared on a large scale via the dry stretching method [5,6,7]. However, polymer separators have drawbacks, including high crystallinity, low polarity, low surface energy, and poor wettability with the electrolyte, leading to electrolyte leakage. Furthermore, the stretching process results in low porosity, reducing the absorption rate of the electrolyte, impeding lithium-ion transport, and negatively impacting the battery’s electrochemical performance [8]. Additionally, the thermal stability of polymer separators is limited. In the event of battery overcharge or abuse resulting in thermal runaway, the separator may fail to shut down the reaction in time, leading to extensive contact between positive and negative materials and a significant threat to battery safety.

The functionalized composite separator, which modifies the polymer separator with advanced material as an interlayer between the conventional separator and electrode, has been developed as a promising strategy to provide excellent insulation, superior thermal/mechanical strength, and an appropriate self-shutdown function. Meanwhile, the higher wettability offers high-speed ion transfer channels, and the active material and functional groups loaded on the surface of the separator promote electron storage, which improves the specific capacity of batteries. Particularly, a rationally designed functional separator can also inhibit the severe lithium dendrite growth in the lithium-metal battery and the shuttle effect of polysulfides in the lithium-sulfur battery. As the star material over recent years, graphene-based materials (such as graphene, graphene oxide (GO), reduced graphene oxide (rGO), and graphene-related composite) are widely used in the field of electrode materials, and the application of them to the separator has received various attempts [9,10]. Separators functionalized with graphene-based materials have apparent advantages, such as great chemical stability, excellent mechanical properties, and outstanding electrical/thermal conductivity, derived from monoatomic layer structure based on the sp^2^ hybridized hexatomic carbon ring of graphene. Meanwhile, the controllable surface functional groups enable more specific properties to be achieved, for instance, the effective catalysis action on redox reactions and the adsorption effect of Li^+^, which significantly improves energy density and cycle stability of batteries while expediting the Li^+^ transfer and electrochemical reaction kinetics [11]. This review summarizes the preparation methods of advanced graphene-based materials reported in recent years, including typical processes such as chemical vapor deposition (CVD) and pyrolysis, and it discusses their applications as functionalized separator materials in lithium-ion batteries, lithium-metal batteries, and lithium-sulfur batteries. Finally, we look forward to the development of next-generation functionalized separator materials and point out the direction for industrialization. Graphical abstract for the current preparation method and application of graphene-based materials for the functionalized separator is shown in Figure 1.

## 2. Preparation Methods of Graphene-Based Materials

Graphene, a two-dimensional carbon material arranged in a honeycomb structure by hybrid sp2 C atoms, has attracted significant attention from researchers for its unique properties. Since the proposal of micromechanical cleavage in 2004 [12], a multitude of methods for graphene preparation have been reported, the most common being methods such as CVD, pyrolysis, exfoliation, etc. Here, we will focus on some representative strategies that have been widely applied in laboratory research and industrial production.

### 2.1. CVD

CVD is appropriate for preparing graphene-based materials with a regular and controlled structure, which can ensure the integrity of the lattice of the graphene sheets forming the structure with minimal defects. When the products are prepared by CVD, the morphology relies on the template material, while the structural stability depends on the type and concentration of the carbon source employed. The use of metals, structural oxides, salts, and combinatorial templates as hard templates for the preparation of graphene-based materials by CVD has been widely reported.

In the utilization of metal temples, Ni and Cu templates have been widely studied for large-scale, high-quality graphene preparation. Chen et al. achieved CVD growth of graphene foam (GF) on a Ni foam template with CH_4_ as the carbon source and polymethyl methacrylate (PMMA) as the support. After etching off Ni with hydrochloric acid and removing PMMA with acetone, a highly conductive graphene foam was obtained (Figure 2A) [13]. An analogous method was used by Ito et al. to prepare nanoporous nickel (np-Ni) templates to manufacture small-bore graphene [14]. By imprinting the Rosette-type Ni and Si substrates as templates, network graphene films can be prepared [15]; graphene fibers can be made with Cu wires [16], and foams can be prepared with Ni or Cu foams (Figure 2B) [17]. Graphene with an internal cross-linking structure can be obtained by controlling the number of layers, size, and assembly angle of multilayer Ni stencils [18]. Li et al. prepared large-area single-layer graphene sheets with excellent quality using benzene as the carbon source and electropolished Cu foil as the template at a low temperature [19]. Kong reported the preparation of a large-area continuous graphene film using polycrystalline Ni films. The Ni films were electron-beam evaporated onto SiO_2_/Si substrates and annealed before CVD synthesis [20]. In the same year, Ruoff et al. used a mixture of methane and hydrogen to achieve the growth of graphene films on Cu substrates at 1000 °C. The size of these films reached the centimeter scale, most of which were single-layer structures [21]. For the advantages of low cost, availability of easy etching methods, and controllability with the guide of the Fe–C phase diagram, Xue et al. synthesized few-layer graphene on a Fe foil template coated with PMMA. The Fe foil was removed by an aqueous solution of FeNO_3_/FeCl_3_ (Figure 2C) [22].

Despite the progress achieved by single metal substrates, as previously mentioned, the great challenge for them is controlling the layer number and uniformity of graphene. Hence, the preparation of graphene by CVD on alloy substrates has rapidly developed due to the advantage of multiple metal templates. Ruoff reported the CVD growth of graphene on “90–10” Cu-Ni alloy, where a higher temperature, longer deposition time, and slower cooling rate helped to increase graphene coverage on Cu-Ni alloy foil (Figure 2D) [23]. Wang successfully realized the control of the graphene layer thickness by adjusting the proportion of Ni in the Cu-Ni alloy [24]. As shown in Figure 2E, Dai et al. designed a combined Ni-Mo system with a layer of Ni deposited onto Mo. In this structure, all of the excess carbon atoms were captured by Mo in the form of MoC, which overcomes the limitation of uneven growth of graphene layers on Ni [25]. In addition, Co-Cu [26], Au-Ni [27], Ni-Ge [28], and other binary alloys have also been reported as excellent systems for controlling the number of graphene layers. These alloy systems are typically composed of a carbon-soluble surface material (Ni or Co coating) and a carbon-insoluble substrate (Ge or Cu foil). Combining the advantages of both, high-quality graphene with controllable layers can be prepared.

In addition, structural oxides have also been studied as a template. Jiang et al. developed graphene nanocages from the in situ decomposition of magnesium oxide templates with alkaline magnesium carbonate [29]. As reported by Büchner, graphene could be grown bottom-up on MgO at a temperature as low as 325 °C using cyclohexane as the carbon source (Figure 2F) [30]. Graphene can also be prepared by CVD using porous metal oxide as a template. Thus, a large number of mesopores can be obtained through the self-assembled macroporous framework of the template, and the metal oxide can be used as a catalyst to produce micropores. CaO has a similar structure to MgO and can also be used as a template for the preparation of graphene. Tang et al. reported the CVD growth of porous graphene materials on CaO templates at 950 °C by feeding CH_4_ (Figure 2G) [31]. Shi et al. used scallops, conch, starfish, and other biomass rich in CaCO_3_ to prepare porous CaO substrates at high temperatures. By controlling the morphology of CaO substrates, the microstructure and corresponding properties of the graphene foam were changed [32]. Similarly, strategies that adopted ZnO [33] and layered double oxide [34,35] as the template have also been reported. 

SiO_2_ is a suitable template for the nucleation and growth of graphene as well. Chen et al. reported the preparation of graphene by using kieselguhr as a template, which inherited the naturally curved surface of kieselguhr and overcame the problem of interlayer stacking [36]. Bi et al. reported graphene with a tetrahedral-connected cell structure via SiO_2_ aerogel templates, which showed a tubular network of several graphene layers with hollow tetrahedral connections (Figure 2H) [37]. Xu et al. presented the fabrication of free-form graphene foam by a combination of 3D printing and CVD. The porous SiO_2_ templates were prepared by digital photo processing methods and polymerization of resins. The stencil porosity was governed by a gluing and burning process, whose product is inherited from an interconnected network with excellent electrical conductivity and mechanical performance [38]. Wei et al. proposed a plasma-enhanced CVD scheme to prepare a highly oriented pyrolytic graphite (HOPG) with H_2_ plasma known to etch graphene from the edges. HOPG has been grown directly on SiO_2_/Si substrate without a catalyst at a temperature as low as 300 °C when using C_2_H_4_ as the carbon source [39]. 

**Figure 2 materials-16-04449-f002:**
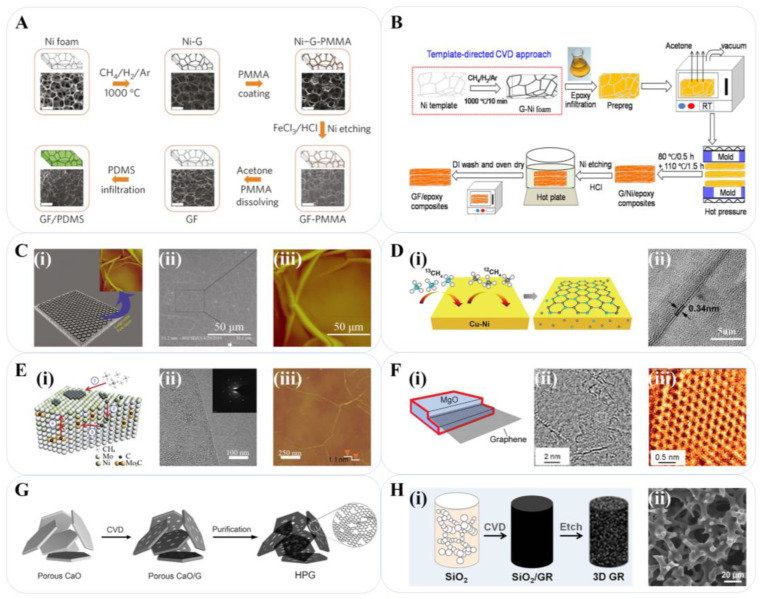
CVD for graphene preparation. (**A**) Synthesis of a GF and integration with PDMS [13]. Reproduced with permission: Copyright 2011, Springer Nature. (**B**) Flowchart for the preparation of GF/epoxy composites [17]. Reproduced with permission: Copyright 2014, American Chemical Society. (**C**) (**i**) Image of a CVD-grown graphene film on iron foil and (inset) AFM image of graphene; (**ii**) SEM image of the graphene; (**iii**) AFM image of graphene on an iron substrate which reveals its surface morphology [22]. Reproduced with permission: Copyright 2011, Springer Nature. (**D**) (**i**) Schematic diagrams of the growth process of carbon isotope labeled graphene; (**ii**) High-resolution TEM image of the graphene [23]. Reproduced with permission: Copyright 2012, American Chemical Society. (**E**) (**i**) CVD growth of single-layer graphene on Ni-Mo substrate, and the numbers from 1 to 4 represent the elementary steps in the Ni–Mo-CVD process; (**ii**) TEM image and the selected area electron diffraction pattern (inset) reveal the nice crystallinity of single-layer graphene grown on the Ni-Mo alloy; (**iii**) AFM image of the grown graphene [25]. Reproduced with permission: Copyright 2011, Springer Nature. (**F**) (**i**) Image of a CVD-grown graphene film on MgO; (**ii**) Graphene island on the surface of a MgO crystal; (**iii**) Magnified region from the box in panel C highlighting graphene structure [30]. Reproduced with permission: Copyright 2010, American Chemical Society. (**G**) Schematic illustration of hierarchical porous graphene obtained by CVD growth on CaO templates [31]. Reproduced with permission: Copyright 2015, John Wiley and Sons. (**H**) (**i**) Schematic synthesis of graphene nanoporous monolith using mesoporous silica template; (**ii**) Low magnification field emission scanning electron microscopy (FESEM) image of graphene nanoporous monolith [37]. Reproduced with permission: Copyright 2016, American Chemical Society.

### 2.2. Pyrolysis

On an industrial scale, pyrolysis using liquid or solid carbon sources can be more efficient and economical than the CVD template method, which requires a gaseous carbon source and more sophisticated equipment, or the solution method, which necessitates a solvent to disperse graphene oxide (GO) flakes to obtain large amounts of graphene. However, pore size cannot be precisely controlled in pyrolysis. To address this issue, the addition of in situ pore-building additives at high temperatures has been explored as a competitive strategy. For instance, Zhang et al. developed a simple method to produce a fluorescent graphene layer by pyrolyzing amide salt [40]. Xu et al. prepared a high-performance magnetic graphene aerogel doped with nickel nanoparticles (GA@Ni) through a simple one-pot hydrothermal route and in situ pyrolysis process. The product revealed a low density and microwave absorption (Figure 3A) [41]. Pejman et al. fabricated vertically structured graphene sheets with 2–10 layers by bombarding polyethylene powder with a microwave plasma emission device and demonstrated that increasing the calcination temperature in an inert or low-oxygen atmosphere enhanced the crystallization of graphene sheets [42].

Pore-forming agents and shape-forming agents (SFA) are commonly employed in the pyrolysis process to fabricate graphene materials with diverse structures while precisely controlling the pore size and distribution [43]. For instance, Zhao et al. generated copper templates in situ during the pyrolysis process by mixing Cu_2_(OH)_2_CO_3_ with polymethyl methacrylate (PMMA) and subsequently induced PMMA to yield few-layer graphene [44]. Bo et al. prepared porous iron-nitrogen functionalized graphene (Fe/N/GR) utilizing ZnO as both the adsorbent and pore-forming agent for GO. The pyrolysis of ZnO/GO in the presence of a nitrogenous precursor and Fe source produced porous Fe/N/GR, in which the Fe species and ZnO were removed, as shown in Figure 3B [45]. Yoon et al. employed nickel nanoparticles (Ni-NPs) as the SFA material to synthesize multilayer graphene balls (GBs) by Ni-induced carbon separation [46]. Li et al. reported the bottom-up construction of ordered mesoporous graphene framework (MGF) thin films in which the oleylamine ligands were pyrolyzed into amorphous carbon shells in the presence of Fe_3_O_4_ nanocrystal (Fe_3_O_4_ NCs) superlattice films and further transferred into MGF after removing Fe_3_O_4_ NCs [47].

Considering the previously mentioned relatively complicated treatments for metal or metal oxide after pyrolysis, salt SFAs have attracted the attention of researchers due to their low cost and high solubility in water. Cui et al. used Na_2_CO_3_ as the skeleton to build the framework structure in which the fumaric acid particles were directly converted into graphene frameworks (GFs) through rapid decomposition [48]. Chen et al. introduced NaCl into the pyrolysis of sodium polyacrylate precursor, which led to the introduction of nitrogen and sulfur atoms into the interlayer of carbon material to prepare nitrogen-sulfur co-doped porous carbon (NSPC) through a simple liquid-solid absorption method (Figure 3C) [49]. Zhao et al. reported a reverse-solvent method strategy to prepare the ferric phthalocyanine (FePc)/KCl precursor. Graphene loaded with Fe particles was formed by surface pyrolysis of FePc (Figure 3D) [50]. Wang et al. studied the strategy of ferric chloride hexahydrate as the pore-forming agent [51], and muti-templates schemes [52] were also studied.

Self-sacrificing SFAs that can be removed or eliminated from the major material during pyrolysis without subsequent elution have also been explored. Cao et al. proposed a strategy to use self-sacrificing graphitic carbon nitride (g-C_3_N_4_) as the stratification agent and pore-forming agent for graphene preparation [53]. Liu et al. applied this strategy for the preparation of vanadium nitride/nitrogen-doped graphene (VN/NGr) in the two-stage pyrolysis of a mixture including dicyandiamide (DCDA), glucose, and NH_4_VO_3_, as shown in Figure 3E. In the first stage, NH_4_VO_3_ was transformed into VN gradually at 600 °C. Thermal decomposition of DCDA produced layered g-C_3_N_4_, which bound the carbon derived from glucose and VN to its interlayer gaps to form graphene layers and meanwhile limited the aggregation of VN nanoparticles. In the second stage, the g-C_3_N_4_ was completely decomposed at 800 °C. The formed nitrogen was introduced into the graphene framework due to the small atomic size difference between N and C. Finally, VN/NGr nanocomposites were obtained without any post-treatment [54]. The strategy was then studied to prepare Co/N/S Tri-doped graphene [55]. Wang et al. proposed a zinc-assisted solid pyrolysis (ZASP) method to produce block graphene. The authors compressed glucose powder and Zn metal (usually 1:9) into an ingot and directly pyrolyzed it to obtain graphene with a full film and excellent graphitization. In addition to the pore-forming effect of common SFAs, zinc metal introduced a stratification process on large organic matter and solid carbon, leading to the preparation of graphene sheets at high temperatures. Zinc gas can be extruded and evaporated from the carbon material at higher temperatures, thereby eliminating the formed impurities. Zinc gas also promoted the carbonization and graphitization of organic matter, realizing the recycling of metal reagents and avoiding tedious wet processing [56].

Removing redundant SFA completely is a challenging task due to potential, unpredictable side effects, which has led to the extensive study of SFA-free pyrolysis. In 2013, Wang et al. developed a chemical blowing route known as sugar blowing, which uses NH_3_ to produce strutted graphene foam. In the process of heating glucose and NH_4_Cl, the molten syrup gradually polymerizes into melanin, which is blown into a network of polymeric bubbles by NH_3_. The walls of these bubbles become thinner due to blowout, drainage, and chemical elimination reactions, leading to the formation of ultra-thin graphene films in the subsequent high-temperature graphitization process. The resulting self-supporting graphene products possess high conductivity, large SSA, and superior mechanical properties [11]. In 2019, Wang et al. proposed an O_2_-NH_3_ reactive pyrolysis strategy to generate graphene, which transformed cellulose paper into networked carbon paper composed of graphene sheets while avoiding the production of dense carbon fiber/particles [57]. Tour et al. applied flash Joule heating technology to the preparation of gram-scale graphene with a turbine tier arrangement structure between the stacked layers from cheap carbon sources in less than one second. The method has attracted the attention of researchers widely on account of its many advantages, such as superior production efficiency, high purity without a furnace, solvent, or reaction gas, and promise for industrial application [58,59].

Graphene can also be prepared by pyrolyzing the surface of single SiC crystals pretreated by oxidization [60] or H_2_ etching [61], in which the Si atoms are removed by evaporation and graphene grows along the surface structure of SiC. The method is also known as “epitaxial growth”, with SiC as both the carbon source and SFA material [62]. Berger et al. subjected single-crystal 6H-SiC to oxidation or hydrogen etching treatment at 1000 °C to eliminate surface oxides under ultra-high vacuum conditions. The sample was then maintained at a temperature range of 1250–1450 °C for 1–20 min, resulting in graphene with a thickness of 1–2 carbon atoms on the silicon-terminated (0001) face [63]. Pakdehi et al. proposed a polymer-assisted sublimation growth (PASG) technique, in which a polymer adsorbent was adsorbed onto 6H-SiC and 4H-SiC single crystals, followed by a two-step heating process to prepare graphene monolayers with near perfect resistance isotropy [64]. Bao et al. employed a rapid cooling process to treat the pyrolyzed SiC samples, causing the disruption of the atomic bonds between SiC and graphene, the transformation of the buffer layer into quasi-free standing monolayer graphene, and the reduction of electron scattering caused by interfacial phonons. The result could be attributed to the negative thermal expansion coefficient of graphene and the positive expansion coefficient of SiC, which led to the expansion of graphene and contraction of SiC during the cooling process [65].

**Figure 3 materials-16-04449-f003:**
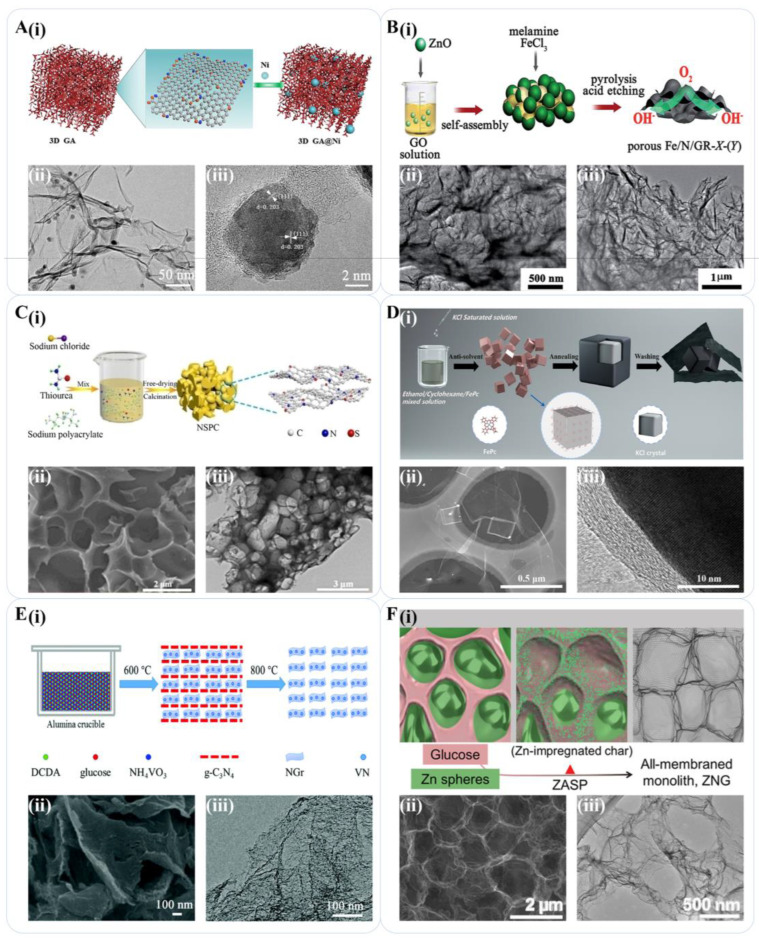
Pyrolysis scheme for graphene preparation. (**A**) (**i**) The structure of GA@Ni hybrids; (**ii**) Representative TEM and (**iii**) High-resolution transmission electron microscopy images of the GA@Ni composites [41]. Reproduced with permission: Copyright 2019, Elsevier. (**B**) (**i**) Pyrolysis method for porous graphene preparation; (**ii**,**iii**) Transmission electron microscopy (TEM) images of porous Fe/N/GR [45]. Reproduced with permission: Copyright 2015, Royal Society of Chemistry. (**C**) (**i**) Schematic diagrams of the synthesis of NSPC; (**ii**) SEM image and (**iii**) TEM image of NSPC [46]. Reproduced with permission: Copyright 2019, John Wiley and Sons. (**D**) (**i**) Schematic diagram of sheet/box Fe@Gr synthesis route; (**ii**) Dark-field scanning transmission electron microscopy of Fe@Gr; (**iii**) High-resolution transmission electron microscopy image of the interface between iron particles and graphene in Fe@Gr-700 [50]. Reproduced with permission: Copyright 2022, Elsevier. (**E**) (**i**) Preparation of VN/NGr by g-C_3_N_4_ strategy; (**ii**) SEM images and (**iii**) low-resolution TEM images of VN/NG [54]. Reproduced with permission: Copyright 2018, Royal Society of Chemistry. (**F**) (**i**) Zinc-induced graphene ZNG synthesis scheme; (**ii**) SEM and (**iii**) TEM of ZNG cells [56]. Reproduced with permission: Copyright 2019, John Wiley and Sons.

### 2.3. Others

In addition to the aforementioned CVD and pyrolysis techniques, various innovative physical and chemical methods have been explored to achieve the reduction of GO, such as arc discharge and exfoliation strategies. These efforts have primarily aimed to control the number of graphene layers, increase yield and purity, minimize defects, and enable the development of devices with superior properties.

#### 2.3.1. Micromechanical Cleavage

Micromechanical cleavage was proposed by Geim et al. in 2004 to prepare high-quality graphene by physically or chemically breaking the van der Waals forces between graphite layers and exfoliating the graphite flakes (Figure 4A) [12]. This method involves etching a groove onto the surface of highly oriented pyrolytic graphite (HOPG) with an oxygen iso-ion beam and pressing it onto a SiO_2_/Si substrate with a photoresist. After roasting, excess graphite flakes are repeatedly peeled off with transparent tape, and the remaining graphite flakes on the Si wafer are soaked in an acetone solution and ultrasonically cleaned. Most of the thicker flakes are removed, leaving graphene flakes less than 10 nm thick. In 2005, Geim et al. used this method to prepare monolayer graphene, demonstrating that monolayer graphene is able to exist independently [66] and successfully prepared suspended planar monolayer graphene with a folded structure supported by metal in 2007 [67]. Due to its low yield, the scheme is only suitable for graphene preparation in the laboratory and not for large-scale production and application.

#### 2.3.2. Liquid Phase Exfoliation

The liquid phase exfoliation (LPE) method has been widely investigated due to its high yield, low defect, low cost, and suitability for large-scale production. Combining the cavitation effect of ultrasound and the intercalation effect of solvent molecules to exfoliate graphite to prepare graphene, i.e., ultrasound-assisted LPE (ULPE), is the most common LPE strategy. Li et al. demonstrated that graphite fragmentation and exfoliation occurred in three different stages: (1) graphite flake rupture and kink band formation; (2) peeling off of thin graphite strips; and (3) exfoliation to thin flakes during ultrasound-assisted LPE in which the kink band-induced exfoliation process was a critical step (Figure 4B) [68]. Gu et al. systematically investigated the effects of three liquid phase systems (i.e., organic solvent system, aqueous surfactant system, and ionic liquid system) on graphene preparation and discussed the exploration of new solvents and improvement measures for the ULPE method [69]. Khan et al. achieved a high yield of monolayer graphene with low basal plane defects. The graphene solution was prepared through mild ultrasonication in N-methylpyrrolidone (NMP) for an extended period [70]. Tung et al. introduced graphene oxide (GO) as a dispersant into ULPE for the first time to prepare graphene nanoflakes (GNFs) with a low oxygen content, few defects, and high electrical conductivity. The GO/graphene complexes formed by wrapping the graphene nanosheets with GO effectively improved the exfoliation rate and dispersion of graphene in an aqueous solution, thus avoiding the restacking of graphene sheets into graphite, and the retained GO in solution could be removed and recycled by centrifugation [71].

Shear-induced LPE has also been investigated for its simplicity of process and feasibility in preparing graphene/polymer composites in situ. Paton et al. demonstrated that large amounts of high-quality, unoxidized graphene suitable for industrial production could be obtained by shear mixing without prior intercalation expansion. They also developed a quantitative model to scale up graphene production using this method [72]. Lynch-Branzoi et al. prepared graphene-enhanced polymer matrix composites (G-PMCs) directly in molten thermoplastic polymers where high-speed shear stress was applied to exfoliate graphite [73]. Based on Tung’s study, Sellathurai et al. prepared graphene nanoplatelets (GNPs) for high-performance supercapacitors by the high-speed shear-induced exfoliation of graphite in an aqueous medium mixed with GO dispersant, then the GO retained in solution could be converted to reduced graphene oxide (rGO) by thermal or chemical reduction [74]. Hadi et al. put forward a nanoparticle-assisted liquid phase exfoliation strategy to prepare graphene with a controlled layer number, which utilized the strong shear forces resulting from the collision of Fe_3_O_4_ particles with graphite particles and intense ultrasonic waves to enhance the exfoliation of graphite [75].

#### 2.3.3. Graphene-Oxide Reduction

The GO reduction method is also reliable for preparing monolayer graphene with high yield and wide application, whose production is usually accompanied by more defects than others. Initially, graphite is oxidized in a strong acid with a potent oxidizing agent to obtain graphite oxide, having various oxygen-containing groups between the layers and carboxyl/hydroxyl groups at the edges. The process weakened the interlayer van der Waals forces and widened the layer distances. Brodie [76] and Standenmaier [77] successively reported the use of KClO_4_ as an oxidizing agent in fuming HNO_3_ and H_2_SO_4_ to oxidize graphite flakes, both of which are time-consuming while producing the toxic and explosive gas ClO_2_ during the reaction process. Hummers proposed an approach using the H_2_SO_4_-NaNO_3_-KMnO_4_ mixture to oxidize graphite, which is the current standard method [78]. External exfoliation, such as ultrasonic exfoliation, can then transform graphite oxide flakes into GO with a single atomic layer thickness (Figure 4C) [79]. Finally, graphene can be obtained by reduction primarily including (1) chemical reduction: the most common scheme, using reducing agents that do not react with water, such as hydrazine [80], hydroquinone [81], etc., to eliminate the oxygen-containing functional group on the surface of graphite oxide for graphene preparation [82]; (2) electrochemical reduction: using linear voltammetric scanning in an electrolytic cell with graphite oxide as the cathode and an inert counter electrode to prepare graphene [83]; and (3) thermal reduction: rapid expansion and exfoliation of graphite oxide occurring at a high temperature and inert atmosphere, while some of the oxygen-containing groups are pyrolyzed and generate CO_2_, ultimately obtaining graphene [84,85]. In addition to changing the temperature, the properties of graphene products can also be controlled by the carbon-oxygen content, the size, and the distribution of pores during the thermal reduction process [86].

#### 2.3.4. Arc Discharge

The direct current arc discharge method is a viable approach for the direct preparation of heteroatom-doped graphene. Rao et al. conducted direct current arc discharge by placing two graphite rods of different sizes in an atmosphere filled with varying ratios of hydrogen and helium mixtures without catalysts and produced mostly 2–4 layer graphene flakes in the inner region of the arc chamber. They explored the optimal conditions for graphene preparation, including current, voltage, and hydrogen pressure [87]. Pham et al. achieved up to 3.5% nitrogen content in the graphene produced via the arc discharge method using GO with solid nitrogen source polyaniline (PANI) as the anode carbon filler, which is significantly higher than in previous related studies (1–1.5%) [88]. Qin et al. systematically investigated the growth mechanism of oligomeric graphene prepared by arc discharge in different buffer gases, including an inert atmosphere of He, an oxidizing atmosphere of O_2_-He, and a reducing atmosphere of H_2_-He. The results showed that the reactivity of the corresponding buffer gas played a crucial role in the generation of few-layered graphene (FG) in the arc discharge method. The graphene growth process was demonstrated in two stages. First, the formation of carbon vapor at the center of the arc and the crystallization and growth of carbon clusters during the diffusion from the center to the side. In the second stage, the reactive gas reacted with the carbon clusters, thus terminating the carbon-centered radical and avoiding the closure of the graphene layer (Figure 4D) [89].

#### 2.3.5. Vacuum Filtration and Liquid-Air Interface Self-Assembly

In order to fabricate graphene-based devices with tailored structures to meet specific application requirements, vacuum filtration, and liquid-air interface self-assembly methods are indispensable. These methods have been widely utilized for the post-treatment of graphene, particularly in the preparation of graphene membranes and composite membranes. Vacuum filtration, for instance, is a prevalent approach for the fabrication of such membranes. Under the effect of vacuum attraction, the flow of graphene-based dispersion generates an orientation force vertical to the filter membrane, leading to the interlocking and parallel assembly of graphene sheets. The density and thickness of the as-filtered membrane can be controlled simply by adjusting the filtration time, concentration, and/or volume of the graphene dispersion [90]. In 2007, Dikin et al. reported a free-standing GO paper via vacuum filtration of GO colloidal dispersion, of which the thickness could be controlled by adjusting the volume of the dispersion solution [91]. Huang et al. reported a graphene oxide-polyethyleneimine hybrid (GPH) film. The graphene oxide-polyethyleneimine (GO-PEI) solution obtained by ultrasound-assisted LPE was filtrated with an acetylcellulose membrane filter under a vacuum condition to prepare a drug-loaded GPH film (Figure 4E) [92]. Wang et al. introduced polystyrene microsphere templates into vacuum filtration for porous graphene preparation [93]. Pham et al. reported a graphene/carbon nanotube (Gr-CNT) composite film via vacuum filtration and successive calcination [94]. On the other hand, the liquid-air interface self-assembly method relies on the orientation force generated during the volatilization process of the dispersion solvent, resulting in the assembly of graphene at the liquid-air interface to form the desired structures (Figure 4F) [95]. Chen et al. reported a uniform wrinkled graphene (WG) grown on the interlayer of an air-ethanol/deionized water solution [96]. Wei et al. reported a free-standing graphene-based film assembled via solvent evaporation of a PVP-stabilized graphene suspension at the liquid-air interface [97].

**Figure 4 materials-16-04449-f004:**
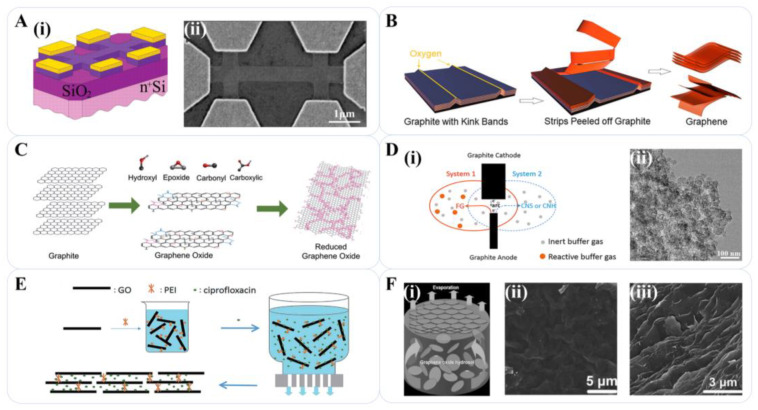
Other methods for graphene preparation. (**A**) (**i**) Schematic view of mechanical exfoliation device; (**ii**) Scanning electron microscope image of one of our experimental devices prepared from FLG [11]. Reproduced with permission: Copyright 2004, The American Association for the Advancement of Science. (**B**) Mechanisms of the LPE process for the production of graphene [68]. Reproduced with permission: Copyright 2020, American Chemical Society. (**C**) Mechanisms of reduction of GO for graphene preparation [79]. Reproduced with permission: Copyright 2012, Elsevier. (**D**) (**i**) Schematic diagram of graphene preparation by arc discharge; (**ii**) TEM images of carbon nanospheres (CNSs) produced in He atmosphere at 73 kPa [89]. Reproduced with permission: Copyright 2016, Elsevier. (**E**) Vacuum filtration process for GPH film preparation [92]. Reproduced with permission: Copyright 2015, Royal Society of Chemistry. (**F**) (**i**) Illustration for GO membrane preparation by liquid-air interface self-assembly; (**ii**,**iii**) SEM images of GO [95]. Reproduced with permission: Copyright 2009, John Wiley and Sons.

## 3. Applications of Graphene-Based Materials for the Separator Functionalization in Lithium-Ion/Metal/Sulfur Batteries

Recently, considerable attention has been given to modification of the separator, exploring solutions for (1) inadequate chemical/mechanical stability, which causes a short cycle life, poor rate performance, shortcuts, and other unwanted consequences for lithium-ion/metal/sulfur batteries; (2) low thermal stability, which leads to unstable electrochemical performance in varying temperatures, melting, and even combustion under high temperatures; (3) poor wettability to the electrolyte, derived from low polarity and the unsuitable pore size of traditional polyolefin separators, which leads to inferior ion transfer kinetics [98,99]; and (4) nonuniform structure distribution, which may bring about a concentration of current, promoting lithium dendrite growth [100]. Due to excellent conductivity, superb thermostability, outstanding mechanical strength, large specific surface area, and abundant active sites for high electrocatalytic activity, graphene-based material is regarded as one of the most promising candidates for separator modification. Here, we introduce the applications of graphene-based materials for the separator in lithium-ion batteries, lithium-metal batteries, and lithium-sulfur batteries.

### 3.1. The Use of Graphene-Based Materials for the Separator of a Lithium-Ion Battery

Due to high energy density and long cycle life, lithium-ion batteries are regarded as the most favorable choice among all secondary batteries. The separator plays a critical role in enhancing electrochemical performance and avoiding some accidents, such as shortcuts, combustion, and even explosion. Graphene-based materials with higher conductivity, better mechanical strength, large specific area, thermal/chemical stability, and suitable pore structure are widely used for separator modification to provide low impedance, large buffer area for electrode volume change, and abundant active sites, which improve the specific capacity, cycle number, and safety. A suitable pore structure and hydrophilic surface functional group offer excellent wettability, which remarkably promotes ion transfer kinetics.

Oxygen-containing functional groups on the surface of GO material are usually used as the catalyst for a redox reaction, while excellent insulation makes GO-based material a promising interlayer between the separator and electrode. In addition, hydrophilic sites on the surface of GO, such as -OH, -O-, and -COOH functional groups, effectively improve the wettability of the separator to the electrolyte, which promotes Li+ transfer [90]. Liu et al. found that the addition of porous GO nanoflakes significantly improved the thermal stability and electrochemical performance of the electrospun polyurethane (PU) separator. Under 170 °C for 1 h, the Celgard film showed large shrinkage, while the shrinkage of the PU@GO separator was only ~20%. The high porosity of the PU@GO separator enabled an extraordinary electrolyte absorption capacity of up to 733%, far beyond that of traditional Celgard separators (~77%). The high electrolyte retention further increased ionic conductivity to 3.73 mS cm^−1^. The reversible discharge capacity of the PU@GO separator after 100 cycles was 154 mAh g^−1^, indicative of 95.6% of the first cycle [101]. Song et al. used an electrospinning technique to prepare a polyimide@graphene oxide (PI@GO) separator in situ with a graphene oxide (GO) nanosheet as the reinforcing agent, which improved the thermal stability, mechanical properties, and ionic conductivity of the PI membrane. When the working temperature rose to 180 °C, the Celgard separator showed a shrinkage rate of up to 90%, while the composite material PI@GO remained unchanged. The Li/LiFePO_4_ battery with the PI@GO separator had a discharge capacity of 160 mAh g^−1^ at 0.2 C and 50 mAh g^−1^ at 5 C [102]. Using polystyrene (PS) nanoparticles as hard templates, Liao et al. designed and prepared a novel kind of GO grafted hyperbranched polyether (GO-g-HBPE) macroporous membrane without any polymer binder. The GO-g-HBPE separator proved to have good wettability with an electrolyte absorption rate as high as 158% and thermal dimensional stability without dimensional change at 200 °C for 0.5 h. The cell using the GO-g-HBPE-based separator finally showed an initial discharge capacity of 160 mAh g^−1^ and kept 84% of that after 200 cycles (Figure 5A) [103].

The introduction of a graphene composite interlayer offers more options for separator design. Zhu et al. mixed GO onto a poly (vinylidene fluoride-co-hexafluoropropylene) PVDF-co-HFP separator and prepared a PVDF-co-HFP/graphene oxide composite membrane with only 0.1% GO content. The oxygen-containing functional group of the GO particles absorbed the liquid electrolyte into the membrane, creating excellent wettability with the electrolyte. The suitable pore structure and electrolyte wettability provided a large number of Li^+^ transfer channels uniformly distributed in the separator. At 0.5 C, the battery assembled with the PVDF-co-HFP/GO separator exhibited a discharge capacity of 160 mAh g^−1^, with only a 5.0% decrease in capacity after 1200 cycles (Figure 5B) [104]. Bu et al. obtained a graphene/polyvinylidene fluoride (G/PVDF-PL) functional layer via coating and post-treatment of the commercial polypropylene separator. Graphene was supported in the coating layer as a skeleton, which is conducive to the absorption and retention of the electrolyte. The membrane with the porous structure on the surface had good wettability with the electrolyte. The liquid absorption rate of the designed composite separator with the G/PVDF-PL functional layer was 326.67%, which was three times higher than that of the polypropylene separator in comparison. The initial capacity of the battery with the G/PVDF-PL separator was 111.3 mAh g^−1^ at 5 C, without an obvious decrease after 600 cycles (Figure 5C) [105]. The above relevant literature information is summarized in Table 1.

### 3.2. The Use of Graphene-Based Materials for the Separator of a Lithium-Metal Battery

Compared with the previously discussed basic lithium-ion batteries, lithium-metal batteries have garnered significant attention from researchers for their exceptional energy density and high voltage. However, the formation of metal dendrites during the electrode process can react with the electrolyte, leading to a rapid decay in battery cycle life, a substantial reduction in coulomb efficiency and energy density, as well as deformation and melting of the separator due to the puncturing and thermal effects of lithium dendrites. These phenomena can ultimately cause irreversible capacity loss, internal short circuits, and even explosions [106,107,108]. In order to solve these problems, research on electrodes [109,110,111], electrolytes [112,113,114,115,116], and interfaces [117,118,119] has been reported extensively. Importantly, separator materials have also been studied recently, which mainly focus on the suppression effect of lithium dendrite growth and the realization of higher conductivity, better mechanical strength, and thermal stability [120,121,122]. However, commercial polyethylene (PE), polypropylene (PP), and glass fiber separators face significant challenges in meeting these requirements. Consequently, numerous modification methods have been proposed to overcome these bottlenecks.

Attributed to excellent electrical and mechanical properties, graphene or graphene-based composite interlayers have garnered significant attention as coating materials for conventional separators in lithium-metal batteries to provide additional electron transfer paths, increase electron storage and conductivity, reduce the local current density of the positive electrode, and buffer the volume change of the electrode material during the discharge/charge process. Han et al. established an electrochemical lithium deposition model matching the heat transfer model and explored the time-dependent variation of heat generation powers under different deposition overpotential conditions. They demonstrated the existence of local hot spots at the tip of the protuberance area of inhomogeneous Li growth, where the nonuniform distribution of electric field and large potential difference were detected, which concentrated Li-ion flux and caused further growth of Li dendrites (also called “tip effect”). They also produced a polypropylene separator (graphene@PP) covered by a graphene stack for lithium-metal batteries. The graphene side, as in situ thermal dispersion medium, was faced with the lithium metal, which eliminated local heat accumulation around irregular lithium deposition and inhibited further dendrite growth. The Li-Cu cell with graphene@PP as a separator achieved a coulomb efficiency of more than 95% after 250 cycles at 1 mA cm^−2^ [100]. Zhang et al. demonstrated that a graphene-coated separator (G@PP) was able to obstruct Mn ions by adsorption for lithium protection in lithium-metal batteries with an Mn-based (Ni, Co, Mn, or their combination, denoted as NCM) cathode. There were no obvious Li dendrites in Li|NCM523 batteries with a G@PP separator after 50 cycles [123].

The introduction of heteroatoms into the graphene layer is an effective way to provide abundant reactive sites, improve the reaction kinetics, and strengthen the interaction with Li^+^, which contributes to higher specific capacity and cycle stability for Li-metal batteries with a graphene-based separator. Shin et al. prepared nitrogen-sulfur co-doped graphene nanosheet (NSG) dispersions through a GO reduction method, which was coated on one side of a polyethylene separator by the vacuum infiltration method. The NSG on the separator was demonstrated to be effective in inhibiting dendrite growth, promoting the uniform distribution of Li^+^ above the lithium-metal surface, and further avoiding massive, localized lithium precipitation at the electrode. The ultra-thin layer of NSG nanosheets also bolstered the dimensional stability of the polymer separator at high temperatures. In addition, the electrostatic attraction between the lone pair electrons of N/S atoms and the lithium metal enhanced the interfacial interaction between the separator and Li anode, releasing the surface tension of lithium metal. Meanwhile, the NSG was used as a physical barrier and relied on excellent mechanical properties, which also inhibited the initiation of lithium dendrite growth. At a 0.5 C rate, the Li/LiNi_0.8_Co_0.15_Al_0.05_O_2_ cell with PE/NSG as separator delivered an initial discharge specific capacity of 200 mAh g^−1^ and retained 85% of that after 240 cycles, and the impedance kept a low value during 200 cycles, indicating remarkable electrochemical stability and reaction kinetics properties (Figure 6A) [124]. Ye et al. reported a novel type of metal/graphene oxide (M/GO) battery in which M was used as the anode while GO acted as both the cathode and separator. The insulating nature of GO avoided occurring a short circuit of the battery. The discharging capacities of the Li/GO battery reached up to 245.6 mAh g^−1^, 239.7 mAh g^−1^, 227.5 mAh g^−1^, and 210.5 mAh g^−1^ at 0.5 C, 1 C, 2 C, and 5 C respectively (1 C = 175 mA g^−1^) [125]. Gong et al. reported an F-doped reduced graphene oxide fiber (rGOF) attached to an aramid paper separator (rGOF-A). During solid electrolyte interphase (SEI) layer formation, the decomposition of the LiPF_6_ electrolyte produced F^−^ anions, which formed C-F bonds on the rGOF surface, thereby facilitating F doping and generating LiF-stabilized SEI between the Li-metal anode and separator. After 1000 cycles, Li-metal batteries with the rGOF-A reached 98.06% (5 C-rate), 89.80% (10 C-rate), and 79.91% (20 C-rate) of the initial specific capacity [126].

Moreover, composite interlayers incorporating graphene with/without heteroatom doping and other materials have also been reported to realize the multi-functionalization and higher performance of graphene-based separator materials, offering more possibilities for designation. Kim et al. reported a multifunctional trilayer film of polydopamine-graphene-carboxymethylcellulose (PDA/Gr-CMC) deposited on a PP separator. The PDA coating conferred hydrophilicity to the PE separator, which enhanced its wettability with the electrolyte and promoted the uniform distribution of Li-ion flux on the lithium-metal surface. Meanwhile, the carboxymethyl cellulose (CMC) aqueous binder also improved the hydrophilicity of the graphene layer. The Li/LiFePO_4_ (LFP) full cell with PDA/Gr-CMC as the separator revealed a specific capacity close to 100% of the initial discharge capacity after 500 cycles [127]. Rodriguez et al. designed a BN/graphene bilayer separator (BN_x_Gr_y_/PP) which exhibited notably higher wettability with the electrolyte than the bare PP. The BN/Gr layer reduced the local current density of the PP separator during Li plating/stripping, improved the thermal conductivity and mechanical stability, and reduced the impedance of the cell, which resulted in better electrochemical performance. SEM characterization results showed that Li^+^ was deposited in a smoother way, leaving a flat surface without large Li protuberances or sharp Li dendrites when compared to the bare PP film, where the massive Li protuberances (>30 μm) are indicated with red arrows (Figure 6B) [128]. Zhang et al. proposed an environment-friendly transition metal nitride/nitrogen-doped reduced graphene oxide (tMN/N-rGO) multifunctional layer strategy, which prepared the PP separator modified by a VN@N-rGO multifunctional layer. The synergistic effect of N-rGO and lithophilic VN notably improved the electrolyte wettability and mechanical stability of the PP separator for lithium-metal batteries, which maintained 94.9% of their initial discharge capacity after 100 cycles [129]. Afterward, they prepared a protective shield layer consisting of Fe_3_N nanoparticles enfolded by N-rGO (Fe_3_N@NG) for the modification of the PP separator. In the Fe_3_N@NG protective layer, N-rGO was able to buffer the volume change of the lithium-metal electrode, improve the reaction kinetics and cycle life of lithium-metal batteries, and maintain the shape of the separator at different temperatures, owing to the high mechanical strength, excellent thermal stability, and high nitrogen doping content of N-rGO. At the same time, the chemically stable Fe_3_N nanoparticles provided abundant deposition sites for uniform lithium deposition, reduced the local current density applied to the anode electrode, and generated a durable solid electrolyte film. The Fe_3_N@NG/PP separator testified a maximum electrolyte uptake of 114.3% and excellent thermal stability (almost without deformation, even at 150 °C). Li||Li symmetric cells constructed by the Fe_3_N@NG/PP separator revealed an ultra-long cycle stability of more than 2300 h at a current density of 5 mA cm^−2^ [130]. 

In addition, a scheme using W_2_N_3_ as the transition metal nitride was also investigated [131]. Yang et al. reported a reduced GO/Li-Al layered bimetallic hydroxide interlayer prepared by a hydrothermal method and coated on a PP separator (rGO/Li-Al-LDH@PP). This separator displayed an exceptionally high electrolyte uptake of 232.64%. Furthermore, the rGO/Li-Al-LDH@PP separator maintained its shape with only slight shrinkage at 180 °C, while the PP separator melted almost completely at 160 °C. The polarization voltage of the Li||Li symmetric cell with the rGO/Li-Al-LDH@PP separator was nearly constant after cycling for 1100 h at a current density of 1.0 mA cm^−2^ and cut-off capacity of 0.5 mAh cm^−2^ [132]. Chen et al. reported a holey graphene oxide@electrospun polyacrylonitrile (HGO-PAN) composite separator for a lithium-metal battery that maintained a discharge capacity of 135.7 mAh g^−1^ at 2 C after 900 cycles, which is 22.3% higher than that of a commercial separator-based cell [133]. Li et al. reported porous polyacrylamide-grafted graphene oxide bilayer composite (GO-g-PAM) separators for Li metal batteries, which showed an average coulombic efficiency of 98% over 150 cycles under 1 mA cm^−2^ and more than 1900 h cycling with a cycling capacity of 5 mAh cm^−2^ at 20 mA cm^−2^ in symmetric Li||Li cells (Figure 6C) [134]. The above relevant literature information is summarized in Table 2.

**Figure 6 materials-16-04449-f006:**
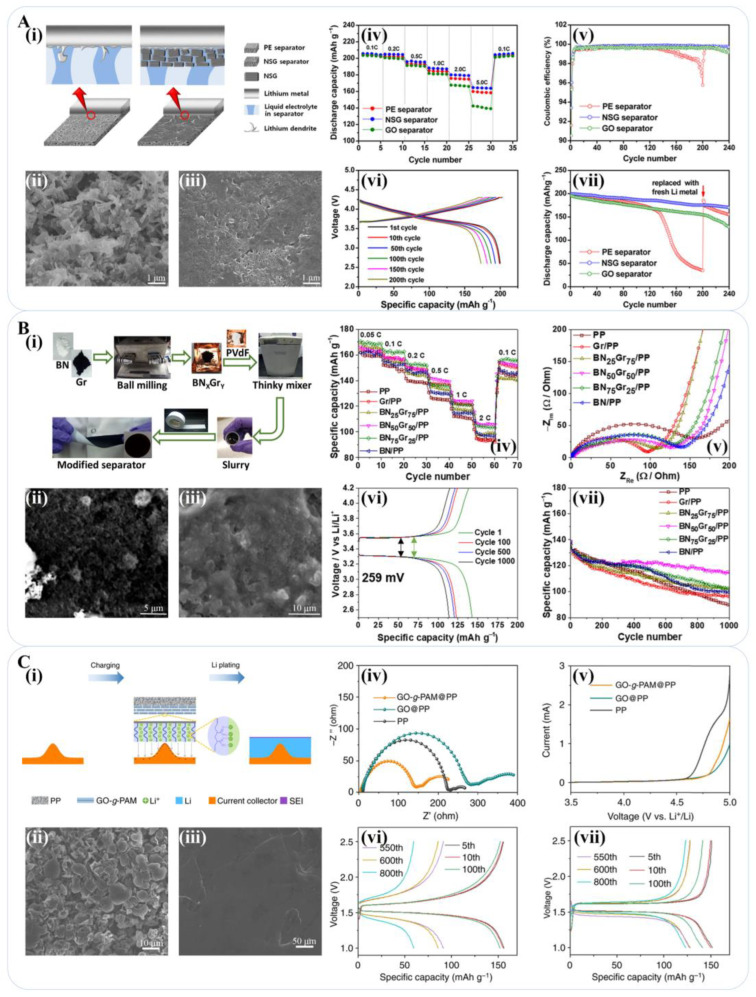
Graphene-based materials for the separator of lithium-metal batteries. (**A**) (**i**) Suppression effect of NSG-coated PE separator on lithium dendrites: PE separator (left) and NSG-coated PE separator (right); (**ii**) SEM images of lithium electrodes after 200 cycles with (**iii**) PE and (**iv**) NSG separators; (**v**) Charge/discharge curves of the lithium-metal batteries with NSG separators after different cycle numbers; (**vi**) Rate capability and (**vii**) discharge capacities of the lithium-metal batteries with different separators [124]. Reproduced with permission: Copyright 2015, American Chemical Society. (**B**) (**i**) Procedure followed to prepare the bilayer BN_x_Gr_y_/PP separator; SEM image of the Li dendrites after 100 cycles using (**ii**) bare PP membrane, and (**iii**) separator modified with BN_50_Gr_50_; (**iv**) rate capability, (**v**) EIS information, (**vi**) Charge/discharge curves of the lithium-metal batteries with BN_x_Gr_y_/PP separators after different cycle numbers and (**vii**) cycle performance of different separators [128]. Reproduced with permission: Copyright 2021, Elsevier. (**C**) (**i**) Schematic illustration of the synthesis of GO-g-PAM composites; SEM images of Li metal anodes assembled with (**ii**) PP and (**iii**) GO-g-PAM@PP separators after 100 cycles; (**iv**) Nyquist plots and of symmetric Li|Li cells with different separators; (**v**) LSV curves of asymmetric Li|stainless-steel cells with different separators; Galvanostatic curves of Li|Li_4_Ti_5_O_12_ cells with (**vi**) GO-g-PAM@PP and (**vii**) PP separators at 3 C [134]. Reproduced with permission: Copyright 2019, Springer Nature.

### 3.3. The Use of Graphene-Based Materials for the Separator of a Lithium-Sulfur Battery

With high theoretical specific capacity (1675 mAh g^−1^) and energy density (2600 Wh kg^−1^) [135], lithium-sulfur (Li-S) batteries are considered one of the most promising next-generation batteries. However, several factors are hindering the practical application of Li-S batteries, including large volume change and the low conductivity of sulfur/lithiated products (Li_2_S_n_, *n* ≥ 1), especially the shuttle effect of polysulfide (Li_2_S_n_, *n* > 1) intermediates generated from the incomplete reduction reaction of the cathode, which moves to react with the Li-metal anode causing irreversible loss of active material of anode. The polysulfide’s back to the S cathode tends to form a sluggish interface on the cathode surface, which further hinders the reduction of polysulfides. From an external view beyond the electrochemical performance of Li-S batteries, the shuttle effect reduces conductivity and Li^+^ transfer kinetics, causing serious self-discharge, a sharp decline of capacity, and poor cycle/rate performance. To overcome these issues, it is necessary to limit the sulfur in certain channels and inhibit the diffusion of polysulfides through physical adsorption and/or chemical combination [136,137,138].

Over the past decades, many works have been reported that tackle the problems mentioned above by changing the electrode structure, adjusting the composition of electrolytes, and modifying the separator [139,140,141]. Among these, separator modification is particularly economical. A suitable separator may prevent the transfer of polysulfides and ensure the normal diffusion of other important ions. A traditional PP separator cannot be used in a Li-S battery system because of its large pore size (~100 nm), which is insufficient to prevent the transportation of polysulfides, and the poor wettability to the electrolyte brings about sluggish ion transfer kinetics through the separator. The graphene-based interlayer with a large specific surface area, abundant adsorption sites, and excellent mechanical properties facilitates the adsorption/capture of polysulfides and buffers the volume change of the S electrode. In addition, constructing a suitable pore size and large pore volume and modifying the graphene material with abundant functional groups on the surface are critical for superior wettability and deposition of polysulfides.

In recent studies, Zhai et al. produced porous graphene (PG) by the CVD method and coated it on the PP separator. This composite layer effectively captured the polysulfide production due to the high specific surface area of PG and the chemical adsorption of PVP, while the high conductivity of PG facilitated the activation of the sluggish interface between the separator and cathode. The Li-S battery with the PG separator showed an initial discharge capacity of 1165 mAh g^−1^ (equivalent to 69.7% sulfur utilization) and retained a reversible capacity of 877 mAh g^−1^ after 150 cycles, with a current density of 0.5 C (Figure 7A) [142]. Ma et al. coated a Celgard 2320 separator with rGO through vacuum filtration. The rGO straddled on the surface of the separator improved the strength of the modification layer, while others inserted into the coating layer increased the conductivity. At 0.2 C after 200 cycles, the reversible specific capacity of the Li-S battery with the rGO-modified separator reached 762.4 mAh g^−1^ with an average coulombic efficiency of 82.7% [143]. Peng et al. reported a Janus separator with an asymmetrical two-face structure constructed by a mesoporous cellular graphene framework (CGF)/polypropylene separator for a Li-S battery. The CGF-modified separator achieved a reversible capacity of up to 800 mAh g^−1^ after 250 cycles at 0.2 C (1.0 C = 1675 mA g^−1^) [144]. Zhou et al. proposed a flexible integrated structure of sulfur and graphene on a PP separator (S-G@PP) for Li–S batteries in which the graphene membrane was coated directly on the PP separator as an internal current collector to support electrode materials with 70 wt% pure sulfur. After 30 cycles at 0.75 A g^−1^, the flexible Li-S battery with the S-G@PP separator retained a capacity of 722 mA h g^−1^ with a Coulombic efficiency of 98% [145].

Weak adsorption of non-polar bare graphene on polar polysulfide hinders the achievement of higher performance by the separator [146,147]. To address this issue, heteroatoms, such as N, P, and S, can be doped onto the graphene surface, forming polar bonds and strengthening the interaction force with polysulfide to inhibit the shuttle effect [147], while the introduction of abundant defects effectively amplified the specific capacity and improved the ion transfer kinetics [148,149,150,151]. Han et al. used H_3_BO_3_/NH_2_CN for GO reduction to obtain B-doped and N-doped graphene and coated them on the separator, respectively, for comparison. The battery with a nitrogen-doped graphene composite separator maintained a specific capacity of 430 mAh g^−1^ after 400 cycles at 0.2 C, delivering better cycling stability [148]. Qi et al. synthesized a two-dimensional layered nitrogen-doped graphene (NG) composite coating on the separator of a Li-S battery. The N atoms mainly distributed at the edge and inside of the carbon plane showed outstanding absorption of polysulfide through dipole-dipole interaction; meanwhile, a catalyzed redox reaction converted soluble polysulfide intermediate into insoluble Li_2_S. At a rate of 1 C, the initial discharge capacity was 794.6 mAh g^−1^ and remained at 575 mAh g^−1^ after 400 cycles [149]. O-containing functional groups, such as carboxyl, aldehyde, and hydroxyl groups, were demonstrated with a stronger anchor effect on polysulfides than on N-containing functional groups and a superb catalytic effect on the conversion of polysulfides. Jiang used the unique excimer ultraviolet light (EUV) technique to induce the generation of O-containing functional groups on graphene material, which was coated onto the glass fiber separator (EUV/graphene separator) for Li-S batteries. The battery with the EUV/graphene separator had an initial specific capacity of up to 1164 mAh g^−1^ at 0.2 C and remained at 640.5 mAh g^−1^ after 400 cycles [150]. Shaibani et al. coated the S electrode with a high-concentration GO solution as an interlayer between the sulfur electrode and the separator. The study found that GO presented as a columnar liquid crystal state at high concentrations. During the coating process, GO was arranged in an ordered structure in the direction of shear stress, presenting a layered and ordered structure that not only suppressed polysulfide ions effectively but also reduced the bending degree and roughness between layers. Thus, it successfully decreased the transport resistance of lithium ions across the GO layer. The carbon composite separator modified with the GO interlayer in the Li-S battery delivered an initial discharge-specific capacity of 1182 mAh g^−1^. It also maintained 835 mAh g^−1^ after 100 cycles at 0.5 C, showing a satisfying sulfur utilization rate [151].

Expected for the combination of advantages from different materials to build up high-performance separators in Li-S batteries, multilayer composites comprising graphene and other materials have been extensively studied. Zhu et al. employed an electrospinning method to prepare a PAN/GO nanofiber membrane separator, which exhibited excellent performance due to the synergistic effect of the N groups of PAN and the O groups of GO. The initial discharge capacities of the Li-S battery with the PAN/GO separator were 699, 591, and 448 mAh g^−1^ at 0.2, 0.5, and 1 C, respectively, while the open-circuit voltage of the battery was as high as 2.73 V, indicating an exceptional anti-self-discharge capability [152]. Xu et al. reported a high-performance ultrathin light GO layer loaded with Cobalt phthalocyanine (CoPc) on a polypropylene (PP) separator. The initial specific capacity of the battery with the CoPc@GO-PP separator was 1092 mAh g^−1^ and retained 757 mAh g^−1^ at 1 C after 400 cycles and 919 mAh g^−1^ after 250 cycles at 0.5 C (Figure 7C) [153]. Jing et al. prepared a sandwich structure modified separator with nano-scale flower strontium fluoride and graphene (SrF_2_-G), which was filtered onto the PP separator. The graphene layer at the upper and lower layers not only acted as an effective physical barrier to block the shuttle of polysulfides but also enhanced electron transport to improve the reactivity. SrF_2_ in the middle layer strongly adsorbed polysulfides and catalyzed the transformation of polysulfides. At a current density of 0.2 C, the initial specific discharge capacity separator of the Li-S battery assembled with the SrF_2_-G separator was 1188 mAh g^−1^ and maintained at 931 mAh g^−1^ after 200 cycles [154]. Zuo et al. synthesized a composite constructed by an N-doped graphene sheet and N-doped carbon nanotube filled with Ni nanoparticles (Ni@NG-CNTs), which was coated on the PP separator for a Li-S battery. The N-doped nanocage structure integrated with a 1D carbon nanotube and 2D graphene was demonstrated to block polysulfide diffusion, providing abundant Li^+^ transfer channels and catalytic sites for redox reactions. The Li-S batteries with the Ni@NG-CNTs-PP separator kept high reversible specific capacities of 309.1 and 265.0 mAh g^−1^ at 5 and 10 C rates after 500 cycles, respectively. At 0.2, 0.5, 1, and 2 C, the discharge-specific capacities were 1036.7, 934.8, 821.9, and 710.9 mAh g^−1^ for the Ni@NG-CNTs-PP separator, respectively [155]. Li et al. used a hydrothermal reaction to synthesize a CuS/graphene composite as an efficient polysulfide immobilizer onto the pristine separator. The separator also reactivated the accumulated necrotic sulfur-containing species. The Li-S battery with the CuS/graphene-coated separator exhibited an initial discharge capacity of 1302 mAh g^−1^ at 0.2 C, maintaining at 760 mAh g^−1^ after 100 cycles [156]. Zhou et al. coated graphene and sulfur directly onto the PP separator (S-G@PP) for a Li-S battery, which strengthened the adhesion between the sulfur and graphene, lowered the interfacial contact resistance, and improved the energy density and rate performance with the lightweight graphene layer as a current collector and barrier for polysulfides. The Li-S batteries with the S-G@PP separator exhibited 1128, 980, 833, 670, 586, and 512 mAh g^−1^ at 0.75, 1.5, 3, 6, 9, and 12 A g^−1^, respectively [145]. Zhang et al. prepared hierarchical oxygen-doped carbon on the surface of reduced graphene oxide (ODC/rGO) for Li-S battery separator modification. The combination of physical adsorption by hierarchical structure and chemical adsorption by oxygen-containing groups exhibited effective adsorption of the diffusing polysulfides. The ODC/rGO maintained a discharge capacity of 456.5 mA h g^−1^ at 5 C, which is much higher than that (53.6 mAh g^−1^) of the battery with the bare glass fiber separator [157]. Lei et al. coated an rGO/sodium lignosulfonate composite layer directly onto the standard PP separator (rGO@SL/PP) for a Li-S battery; abundant negatively charged sulfonic and dendritic groups effectively hindered PS shuttling while ensuring excellent Li^+^ transport kinetics. The Li-S battery with rGO@SL/PP showed an initial specific capacity of 707 mA h g^−1^ at 2 C, which retained 74% over 1000 cycles [158]. The above relevant literature information is summarized in Table 3.

**Figure 7 materials-16-04449-f007:**
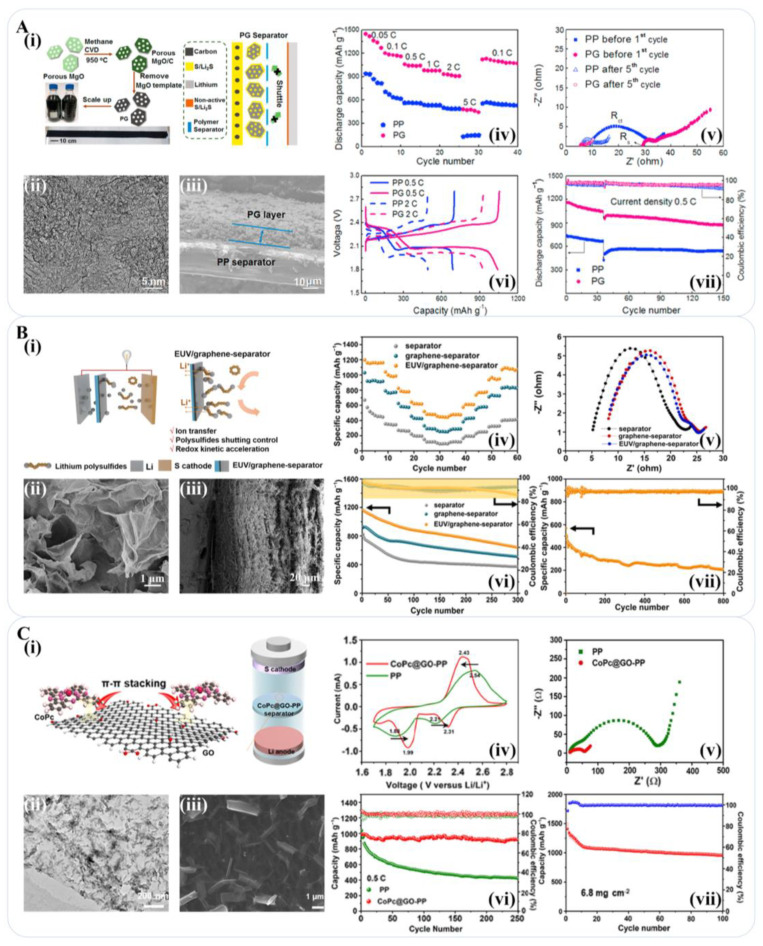
Graphene-based materials for the separator of lithium-sulfur batteries. (**A**) (**i**) Schematic illustration of PG fabrication though methane CVD on porous MgO templates and PG separators in effecting cathode/separator interfaces; (**ii**) TEM image of PG; (**iii**) SEM images of PG separator with a cross-sectional view; (**iv**) Rate performance; (**v**) EIS before the first cycle and after fifth cycle; (**vi**) galvanostatic charge-discharge curves at 0.5 C and 2 C; (**vii**) cycling performance at 0.5 C of the battery composed by PG [142]. Reproduced with permission: Copyright 2017, Elsevier. (**B**) (**i**) Functional Li-S configurations with EUV/graphene separator and polysulfide redox of EUV/graphene separator; (**ii**) SEM image and (**iii**) cross-sectional SEM image of EUV/graphene separator; (**iv**) Rate performance of Li-S battery with different separators at 0.1, 0.2, 0.5, 1, 2, 2.5 C, followed by reduction to 0.1 C; (**v**) Nyquist plot of an uncycled Li–S battery with different separators within a frequency range of 1 MHz to 1 Hz; (**vi**) Cycling performance of the batteries with different separator at 0.2 C; (**vii**) Charge-discharge specific capacity and coulombic efficiencies of the battery with EUV/graphene separator at 2 C over 800 cycles [150]. Reproduced with permission: Copyright 2022, Elsevier. (**C**) (**i**) Schematic of the working mechanism of the CoPc@GO-PP separator for Li-S batteries; (**ii**) TEM image of CoPc@GO; (**iii**) SEM image of the surface of CoPc@GO-PP; (**iv**) CV curves of the battery with the CoPc@GO-PP separator at 0.5 C; (**v**) Electrochemical impedance spectra (EIS) of the batteries with CoPc@GO-PP and PP separators; (**vi**) Cycling performance of the battery with different separator at 0.5 C; (**vii**) Cycling performance of the CoPc@GO-PP separator with a high sulfur loading of 6.8 mg cm^−2^ at 0.2 C [153]. Reproduced with permission: Copyright 2021, American Chemical Society.

## 4. Conclusions and Perspective

Graphene-based materials for separator modification in lithium-ion/metal/sulfur batteries have recently received vast attempts. In this review, we provided a comprehensive summary of the recent advancements in the preparation of graphene-based materials using techniques such as CVD, pyrolysis, and others. We also surveyed their applications as functionalized separator materials in next-generation lithium-ion/metal/sulfur batteries in Table 4.

Generally, functionalized separators utilizing porous graphene-based materials exhibit significant enhancements in conductivity, wettability, and thermal stability due to their large surface area, exceptional chemical stability, excellent mechanical properties, and controllable oxygen-containing surface functional groups. These materials also provide numerous active sites for redox reactions and Li^+^ transfer channels while buffering the volume change of electrodes. Specifically, for lithium-ion batteries, the high wettability and abundant functional groups lead to improved Li^+^ transfer kinetics, superior energy density, and enhanced rate/cycle performance. For lithium-metal batteries, the graphene-based interlayer mitigates local heat accumulation around irregular lithium deposition and inhibits further dendrite growth. Additionally, the outstanding mechanical strength of these materials effectively prevents separator puncturing. Finally, in lithium-sulfur batteries, the graphene-based interlayer can function as an effective sulfur adsorption agent and capture network, as well as an activator for dead sulfur.

However, there are still great challenges for the industrial production of separators assembled with a graphene-based interlayer. Regarding preparation, it is quite challenging to make sufficient integration of low cost, minimal pollution, high yield, and high quality. While micromechanical cleavage can produce graphene sheets of just a few atomic layers with high accuracy, its low production efficiency restricts its application to laboratory settings. CVD is a commonly employed method to produce high-quality single-layer graphene material, but issues, such as the service life of templates and pollution from the etching process, need to be considered for large-scale industrial production. Pyrolysis with SFA encounters difficulties in completely removing the residual SFA, while SFA-free pyrolysis requires even more stringent production conditions. Liquid-phase exfoliation is a low-cost method for preparing large-scale few-layer graphene, but it is challenging to control the layer number and avoid restacking of the graphene layer during extended production times.

In terms of application in separator modification, further research is needed to understand the mechanism of electron/Li^+^ transfer kinetics and the interaction between the graphene-based interlayer and electrolytes/electrodes. Additionally, new composite strategies, rational structure design schemes, and improved film formation and coating methods need to be developed. For example, in situ preparation of an integrated graphene-based interlayer and substrate material may lead to more uniform modification compared to simply coating graphene-based material onto commercial separators, as most studies do at present. Afterward, the self-shutdown function when overheating and shortcuts were rarely mentioned in the present works about separators functionalized with graphene-based materials, which may be a desirable scheme for improved safety. Other potential strategies that are worth further investigation include self-supporting graphene-based interlayers without substrates and the use of graphene-modified separators in gel/solid electrolytes for higher performance in next-generation energy storage devices.

## Figures and Tables

**Figure 1 materials-16-04449-f001:**
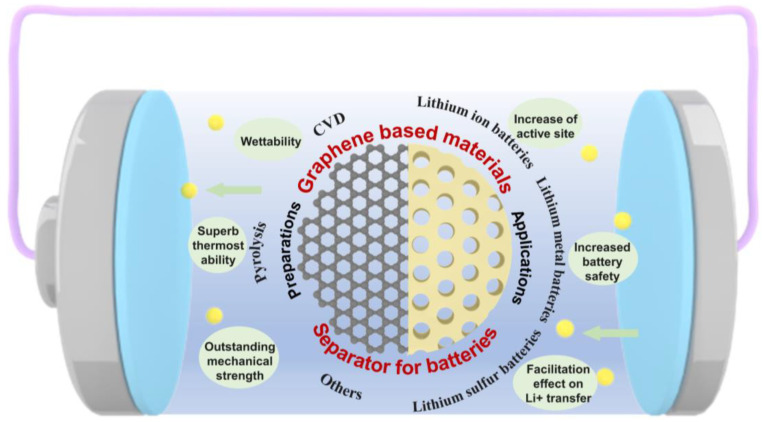
Graphical abstract for the current preparation method and application of graphene-based materials for the separator functionalization of lithium-ion/metal/sulfur batteries.

**Figure 5 materials-16-04449-f005:**
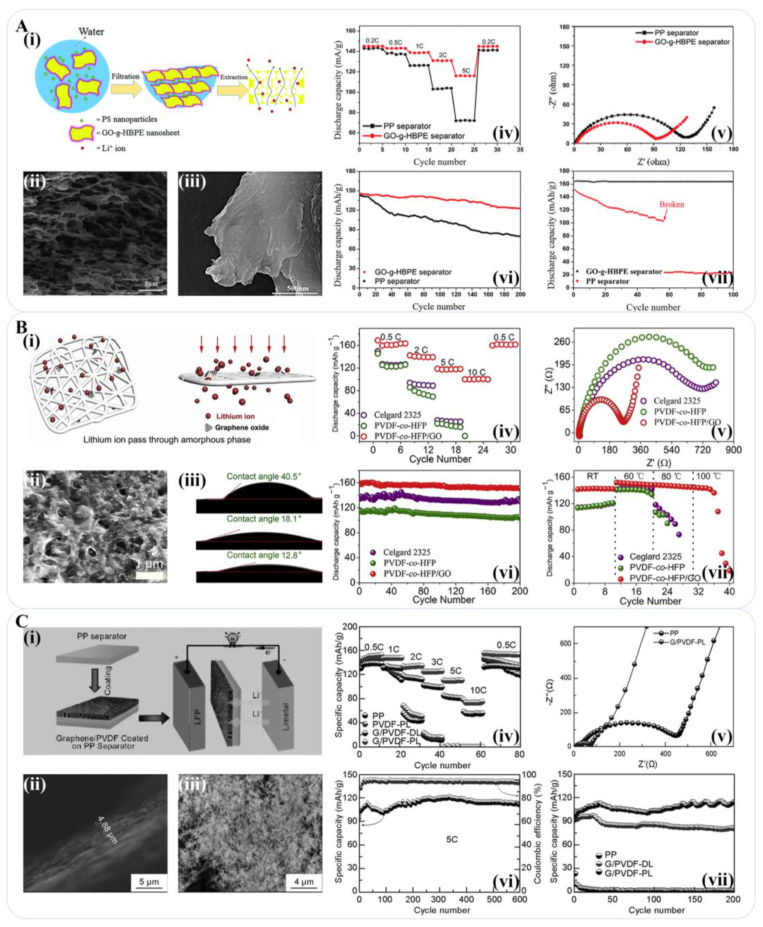
Morphology and electrochemical performance of Graphene-based materials for the separator of lithium-ion batteries. (**A**) (**i**) Mechanism diagram of independent GO-g-HBPE macroporous membrane; the SEM images of (**ii**) high-magnified cross-section of GO-g-HBPE free-standing macro-porous membrane and (**iii**) GO-g-HBPE; Electrochemical performance of the LiFePO4 half cells assembled with PP and GO-g-HBPE separators; Lithium foil served as both the counter and reference electrodes: (**iv**) Rate performance; (**v**) 1st cycle AC impedance spectra of the GO-g-HBPE separator and the PP separator (**vi**) Cycling performance; (**vii**) Charge–discharge performance at an elevated temperature of 80 °C [103]. Reproduced with permission: Copyright 2017, Royal Society of Chemistry. (**B**) (**i**) Schematic diagram of lithium-ion passing through PVDF-co-HFP/GO separator; (**ii**) SEM image of PVDF-co-HFP/GO separator; (**iii**) Contact angle of commercial Celgard 2325; (**iv**) Rate performances at various C rates of LFP/Li cells with different separators; (**v**) Impedance of different separators; (**vi**) Cyclic performances of LFP/Li cells with different separators at 0.5 C; (**vii**) Cyclic performances of LFP/Li cells with PVDF-co-HFP/GO separator at various temperatures [104]. Reproduced with permission: Copyright 2020, Elsevier. (**C**) (**i**) Schematic of the structural and working mechanism of graphene/PVDF composite separator; SEM images of (**ii**) cross-section view of G/PVDF-PL composite separator and (**iii**) surface of G/PVDF-PL composite separator; (**iv**) Rate capability of the battery composed by different separator; (**v**) Electrochemical impedance spectra of PP, G/PVDF-PL separator; (**vi**) The 600 cyclic performance and coulombic efficiency of the battery composed by G/PVDF-PL composite separator; (**vii**) Cyclic performance of the battery composed by different separator at 5 C [105]. Reproduced with permission: Copyright 2017, New Carbon Materials.

**Table 1 materials-16-04449-t001:** A summary of the characteristics of the graphene-based material functionalized separator in Li-ion batteries.

Separator	Thermal Shrinkage (%)	Electrolyte Uptake (%)	Anode /Cathode	Rate Performance	Cycle Performance	Main Functions	Ref.
Celgard 2400	Great shrinkage	77	Li/LiFePO_4_	135, 113, 83, 30 mAh g^−1^ at 0.5, 1, 2, 5 C	138 mAh g^−1^ (0.2 C, after 100 cycles)	(1) Improve wettability; (2) Provide abundant active sites; (3) Ensure excellent thermal/mechanical stability.	[101]
PU@GO	20 (170 °C, 1 h)	733	Li/LiFePO_4_	147, 121, 93, 55 mAh g^−1^ at 0.5, 1, 2, 5 C	154 mAh g^−1^ (0.2 C, after 100 cycles)		[101]
PI@GO	25 (280 °C, 1 h)	-	Li/LiFePO_4_	149, 137, 111, 49 mAh g^−1^ at 0.5, 1, 2, 5 C	-		[102]
GO-g-HBPE	0 (200 °C, 0.5 h)	158	Li/stainless steel	135, 129, 118, 95 mAh g^−1^ at 0.5, 1, 2, 5 C	122 mAh g^−1^ (0.2 C, after 200 cycles)		[103]
PVDF-co-HFP/GO	12 (160 °C, 1 h)	498	Li/LiFePO_4_	164, 139, 118, 100 mAh g^−1^ at 0.5, 2, 5, 10 C	152 mAh g^−1^ (0.2 C, after 200 cycles)		[104]
G/PVDF-PL	-	327	Li/LiFePO_4_	153, 148, 135, 111 mAh g^−1^ at 0.5, 2, 5, 10 C	129 mAh g^−1^ (2 C, after 200 cycles)		[105]

*w* = weight, *v* = volume.

**Table 2 materials-16-04449-t002:** A summary of the characteristics of the graphene-based material functionalized separator in Li-metal batteries.

Separator	Electrolyte	Anode /Cathode	Rate Performance	Cycle Performance	Main Functions	Ref.
PP	1 M LiPF_6_ in DOL/DME (1:1, *v*/*v*) with 2% LiNO_3_	Li/NCM811	197.5, 177, 102, 11 mAh g^−1^ at 0.2, 0.5, 1, 2 C	13 mAh g^−1^ (1 C, after 200 cycles)	(1) Inhibit lithium dendrites growth; (2) Provide abundant active sites; (3) Ensure excellent thermal/mechanical stability.	[100]
graphene/PP	1 M LiPF_6_ in DOL/DME (1:1, *v*/*v*) with 2% LiNO_3_	Li/NCM811	199, 184, 115, 28 mAh g^−1^ at 0.2, 0.5, 1, 2 C	120 mAh g^−1^ (1 C, after 200 cycles)		[100]
PE/NSG	1.15 M LiPF_6_ in EC/DEC (3:7, *v*:*v*)	Li/LiNi_0.8_Co_0.15_Al_0.05_O_2_	195, 187, 178, 162 mAh g^−1^ at 0.5, 1, 2, 5 C	170 mAh g^−1^ (0.5 C, after 240 cycles)		[124]
GO	1 M LiPF_6_ dissolved in EC/DEC/DMC (1/1/1, *w*/*w*/*w*)	Li/GO	245.6, 239.7, 227.5, 210.5 mAh g^−1^ at 0.5, 1, 2, 5 C	-		[125]
rGOF-A	1 M LiPF_6_ in EC/DEC/DMC (1/1/1, *v*/*v*/*v*)	Li/LiFePO_4_	147.49, 100.72, 78.83 mAh g^−1^ at 0.5, 1, 2 C	144.63 mAh g^−1^ (5 C, after 1000 cycles)		[126]
PDA/Gr-CMC	1 M LiPF_6_ dissolved in (EC/EMC/DEM) (1/1/1, *w*/*w*/*w*)	Li/LiFePO_4_	89, 77, 65, 39 mAh g^−1^ at 0.5, 1, 2, 4 C	130.21 mAh g^−1^ (1 C, after 1000 cycles)		[127]
BN_50_Gr_50_/PP	1.0 M LiPF_6_ in EC:DEC	Li/LiFePO_4_	139, 124 106 mAh g^−1^ at 0.5, 1, 2 C	114 mAh g^−1^ (1 C, after 1000 cycles)		[128]
VN@N-rGO/PP	1 M LiPF_6_ in EC/DEC (1/1/1, *v*/*v*)	Li/LiFePO_4_	149.4, 134.1, 111.7, 101.9 mAh g^−1^ at 0.5, 1, 2, 3 C	98.2 mAh g^−1^ (3 C, after 200 cycles)		[129]
Fe_3_N@NG/PP	1 M LiTFSI in EC/DEC (1:1)	Li/LiFePO_4_	149.4, 133.5, 114.6 mAh g^−1^ at 0.5, 1, 2 C	88 mAh g^−1^ (2 C, after 350 cycles)		[130]
WNG/PP	1 M LiPF_6_ in EC/DEC (1/1, *v*/*v*)	Li/LiFePO_4_	135.9, 107.6, 92.4 mAh g^−1^ at 1, 2, 3 C	111.6 mAh g^−1^ (1 C, after 300 cycles)		[131]
HGO-PAN	-	Li/LiFePO_4_	157, 151, 141, 126 mAh g^−1^ at 0.5, 1, 2, 5 C	127 mAh g^−1^ (2 C, after 900 cycles)		[132]
GO-*g*-PAM@PP	1 M LiPF_6_ in EC/DEC (1/1, *w*/*w*)	Li|Li_4_Ti_5_O_12_	-	123.2 mAh g^−1^ (3 C, after 800 cycles)		[133]

**Table 3 materials-16-04449-t003:** A summary of the characteristics of the graphene-based material functionalized separator in Li-sulfur batteries.

Separator	Sulfur Loading (mg cm^−2^)	Rate Performance	Cycle Performance	Main Functions	Ref.
PP	1.8–2.0	564, 533, 490, 141 mAh g^−1^ at 0.2, 0.5, 1, 2 C	540 mAh g^−1^ (0.5 C, after 150 cycles)	(1) Inhibit the transfer of polysulfide; (2) Buffer the volume change of electrodes; (3) Provide abundant active sites; (4) Ensure excellent thermal/mechanical stability.	[142]
PG	1.8–2.0	1038, 975, 903, 440 mAh g^−1^ at 0.2, 0.5, 1, 2 C	877 mAh g^−1^ (0.5 C, after 150 cycles)		[142]
PPy nanotube	2.5–3	195, 187, 178, 162 mAh g^−1^ at 0.5, 1, 2, 5 C	801.6 mAh g^−1^ (0.5 C, after 300 cycles)		[143]
CGF	1.2	1096, 1029, 966 mAh g^−1^ at 0.5 1, 2 C	779 mAh g^−1^ (0.5 C, after 300 cycles)		[144]
S-G@PP	1.5–2.1	1128, 980, 833, 670, 586 mAh g^−1^ at 0.75, 1.5, 3, 6, 9 A g^−1^	663 mAh g^−1^ (1.5 A g^−1^, after 500 cycles)		[145]
N-rGO	4.0	1060, 927, 779 mAh g^−1^ at 0.5, 1, 2 C	758.3 mAh g^−1^ (1 C, after 400 cycles)		[148]
Ni_3_Sn_2_/NG	1.1–1.6	1280.5, 1060.2, 927.5, 778.8 mAh g^−1^ at 0.2, 0.5, 1, 2 C	758.3 mAh g^−1^ (1 C, after 400 cycles)		[149]
EUV/graphene	1.55	824.4, 643.5, 518 and 456.3 mAh g^−1^ at 0.5, 1, 2, 2.5 C	640.5 mAh g^−1^ (0.2 C, after 300 cycles)		[150]
GO membrane	1–1.2	1285, 1256, 870 mAh g^−1^ at 0.2, 0.5, 1 C	835 mAh g^−1^ (0.5 C, after 100 cycles)		[151]
PAN/GO	0.7–1	591, 448, 337 mAh g^−1^ at 0.5 C, 1 C, 2 C	597 mAh g^−1^ (0.2 C, after 100 cycles)		[152]
CoPc@GO-PP	6.8	-	919 mAh g^−1^ (0.5 C, after 250 cycles)		[153]
SrF_2_-G/PP	2.3	1131, 1083, 950, 878 mAh g^−1^ at 0.5, 1, 2, 5 C	811 mAh g^−1^ (0.2 C, after 110 cycles)		[154]
Ni@NG-CNTs-PP	1–1.2	935, 822, 711, 545 mAh g^−1^ at 0.5, 1, 2, 5 C	127 mAh g^−1^ (2 C, after 900 cycles)		[155]
CuS/graphene-coated separator	1.85	999, 864, 701, 523 mAh g^−1^ at 0.5, 1, 2, 5 C	760 mAh g^−1^ (0.2 C, after 100 cycles)		[156]
G@PP	1.5–2.1	980, 833, 670, 586 mAh g^−1^ at 1.5, 3, 6, 9 A g^−1^	663 mAh g^−1^ (1.5 A g^−1^, after 500 cycles)		[145]
ODC/rGO-Coated Separator	0.5	969, 844, 710, 465 mAh g^−1^ at 0.5, 1, 2, 5 C	592 mAh g^−1^ (1 C, after 600 cycles)		[157]
rGO@SL/PP	1.5	701, 603, 490, 465 mAh g^−1^ at 0.05, 0.1, 0.2 C	523 mAh g^−1^ (2 C, after 1000 cycles)		[158]

**Table 4 materials-16-04449-t004:** A summary of the electrochemical performance of the graphene-based material functionalized separator in Li-ion/metal/sulfur batteries.

Separator	Rate Performance	Cycle Performance	Main Functions	Ref.
	Lithium-ion batteries			
Celgard 2400	135, 113, 83, 30 mAh g^−1^ at 0.5, 1, 2, 5 C	138 mAh g^−1^ (0.2 C, after 100 cycles)	(1) Improve wettability; (2) Provide abundant active sites; (3) Ensure excellent thermal/mechanical stability.	[101]
PU@GO	147, 121, 93, 55 mAh g^−1^ at 0.5, 1, 2, 5 C	154 mAh g^−1^ (0.2 C, after 100 cycles)		[101]
PI@GO	149, 137, 111, 49 mAh g^−1^ at 0.5, 1, 2, 5 C	-		[102]
GO-g-HBPE	135, 129, 118, 95 mAh g^−1^ at 0.5, 1, 2, 5 C	122 mAh g^−1^ (0.2 C, after 200 cycles)		[103]
PVDF-co-HFP/GO	164, 139, 118, 100 mAh g^−1^ at 0.5, 2, 5, 10 C	152 mAh g^−1^ (0.2 C, after 200 cycles)		[104]
G/PVDF-PL	153, 148, 135, 111 mAh g^−1^ at 0.5, 2, 5, 10 C	129 mAh g^−1^ (2 C, after 200 cycles)		[105]
	Lithium-metal batteries			
PP	197.5, 177, 102, 11 mAh g^−1^ at 0.2, 0.5, 1, 2 C	13 mAh g^−1^ (1 C, after 200 cycles)	(1) Inhibit lithium dendrites growth; (2) Provide abundant active sites; (3) Ensure excellent thermal/mechanical stability.	[100]
graphene/PP	199, 184, 115, 28 mAh g^−1^ at 0.2, 0.5, 1, 2 C	120 mAh g^−1^ (1 C, after 200 cycles)		[100]
PE/NSG	195, 187, 178, 162 mAh g^−1^ at 0.5, 1, 2, 5 C	170 mAh g^−1^ (0.5 C, after 240 cycles)		[124]
GO	245.6, 239.7, 227.5, 210.5 mAh g^−1^ at 0.5, 1, 2, 5 C	-		[125]
rGOF-A	147.49, 100.72, 78.83 mAh g^−1^ at 0.5, 1, 2 C	144.63 mAh g^−1^ (5 C, after 1000 cycles)		[126]
PDA/Gr-CMC	89, 77, 65, 39 mAh g^−1^ at 0.5, 1, 2, 4 C	130.21 mAh g^−1^ (1 C, after 1000 cycles)		[127]
	Lithium-sulfur batteries			
PP	564, 533, 490, 141 mAh g^−1^ at 0.2, 0.5, 1, 2 C	540 mAh g^−1^ (0.5 C, after 150 cycles)	(1) Inhibit the transfer of polysulfide; (2) Buffer the volume change of electrodes; (3) Provide abundant active sites; (4) Ensure excellent thermal/mechanical stability.	[142]
PG	1038, 975, 903, 440 mAh g^−1^ at 0.2, 0.5, 1, 2 C	877 mAh g^−1^ (0.5 C, after 150 cycles)		[142]
PPy nanotube	195, 187, 178, 162 mAh g^−1^ at 0.5, 1, 2, 5 C	801.6 mAh g^−1^ (0.5 C, after 300 cycles)		[143]
CGF	1096, 1029, 966 mAh g^−1^ at 0.5 1, 2 C	779 mAh g^−1^ (0.5 C, after 300 cycles)		[144]
S-G@PP	1128, 980, 833, 670, 586 mAh g^−1^ at 0.75, 1.5, 3, 6, 9 A g^−1^	663 mAh g^−1^ (1.5 A g^−1^, after 500 cycles)		[145]
N-rGO	1060, 927, 779 mAh g^−1^ at 0.5, 1, 2 C	758.3 mAh g^−1^ (1 C, after 400 cycles)		[148]
Ni_3_Sn_2_/NG	1280.5, 1060.2, 927.5, 778.8 mAh g^−1^ at 0.2, 0.5, 1, 2 C	758.3 mAh g^−1^ (1 C, after 400 cycles)		[149]
EUV/graphene	824.4, 643.5, 518 and 456.3 mAh g^−1^ at 0.5, 1, 2, 2.5 C	640.5 mAh g^−1^ (0.2 C, after 300 cycles)		[150]
GO membrane	1285, 1256, 870 mAh g^−1^ at 0.2, 0.5, 1 C	835 mAh g^−1^ (0.5 C, after 100 cycles)		[151]
PAN/GO	591, 448, 337 mAh g^−1^ at 0.5 C, 1 C, 2 C	597 mAh g^−1^ (0.2 C, after 100 cycles)		[152]
CoPc@GO-PP	-	919 mAh g^−1^ (0.5 C, after 250 cycles)		[153]
SrF_2_-G/PP	1131, 1083, 950, 878 mAh g^−1^ at 0.5, 1, 2, 5 C	811 mAh g^−1^ (0.2 C, after 110 cycles)		[154]
Ni@NG-CNTs-PP	935, 822, 711, 545 mAh g^−1^ at 0.5, 1, 2, 5 C	127 mAh g^−1^ (2 C, after 900 cycles)		[155]
CuS/graphene-coated separator	999, 864, 701, 523 mAh g^−1^ at 0.5, 1, 2, 5 C	760 mAh g^−1^ (0.2 C, after 100 cycles)		[156]
G@PP	980, 833, 670, 586 mAh g^−1^ at 1.5, 3, 6, 9 A g^−1^	663 mAh g^−1^ (1.5 A g^−1^, after 500 cycles)		[145]
ODC/rGO-Coated Separator	969, 844, 710, 465 mAh g^−1^ at 0.5, 1, 2, 5 C	592 mAh g^−1^ (1 C, after 600 cycles)		[157]
rGO@SL/PP	701, 603, 490, 465 mAh g^−1^ at 0.05, 0.1, 0.2 C	523 mAh g^−1^ (2 C, after 1000 cycles)		[158]

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
