# Peer review of "Graphene-Based Materials for the Separator Functionalization of Lithium-Ion/Metal/Sulfur Batteries"

_materials, 2023, doi:10.3390/ma16124449_

Round 1

Reviewer 1 Report

Graphene-based materials for the separator functionalization of lithium ion/metal/sulfur batteries

The purpose of this review is to discuss the application of graphene as separator  for different types of batteries. The topic is very interesting to review and to learn about the latest developments in this specific field. This is a very well written article.

Few comments:

1.     Introduction should at least 3 components/paragraphs:

a.      Introduction to the field - why only focus on lithium batteries? What about metal/sulfur batteries?

b.      problem statement – There are many reviews on graphene for batteries, so why are authors still writing them?

c.      objective and method/approach – very good

2.     What are the authors' summaries on each topic? For instance, there are several papers citing composite layers, is it good or not? (L727)

3.     There is too much simplicity in Figure 1, which does not accurately represent the entire review. As an example, this review reviews a variety of methods for producing graphene. Focus should be placed on separators.

4.     For a better understanding, Figure 1 should show a normal/general battery separator.

5.     The label in the figure is a bit confusing, for example A(a). It is better to write Ai.

6.     L346: Why rGO is better for batteries applications?

7.     L444: How about the application of pure graphene or GO/rGO for separator?

8.     L798: It would be better to include Table 4 in the main text and not in the conclusion.

9.     L665: How about the effect of Li+ diffusion through the separator (graphene based?

10.  L585-636: Too long to fit in a paragraph. A paragraph usually represents one story.

Ok

Author Response

Thank you for reviewing our submission and providing valuable feedback and suggestions. We sincerely appreciate your expertise, and we will make the necessary improvements to the manuscript based on your comments. The corresponding revised manuscript has been uploaded in the attachment.Please find our response to each of the issues and suggestions raised below:

1.Regarding Comment 1:

We sincerely appreciate you raising this comment. We have carefully reviewed the manuscript and made the necessary revisions.

  1. Lithium-ion, lithium metal, and lithium sulfur batteries all fall under the category of lithium batteries, and the issues previously mentioned in the introduction are relevant to all three types. Furthermore, we have incorporated specific discussions regarding lithium metal and lithium sulfur batteries (L66).
  2. The unparalleled properties of graphene, as mentioned in the previous discussion, make it irreplaceable. Therefore, the development of graphene-based functional separators stands as a viable strategy to enhance battery performance.

2.Regarding Comment 2:

Thank you for bringing up this comment. We have reevaluated the relevant evidence and made revisions accordingly. The summaries of each category of material are written at the beginning of the paragraph - what’s the advantages of this material. But there are also weaknesses of each category, so it's hard to give a conclusion about what's good or not, which is also shown on the performance of specific separator materials.

3.Regarding Comment 3:

Thank you for reminding us of this issue. All the figures have been adjusted as required.

4.Regarding Comment 4:

Thank you for reminding us of this issue. All the figures have been adjusted as required.

5.Regarding Comment 5:

Thank you for reminding us of this issue. All the figures have been adjusted as required.

6.Regarding Comment 6:

We appreciate your comment, and we fully agree with your perspective. However,we are discussing the preparation of rGO by LPE method here, without mentioning the application of rGO in batteries.

7.Regarding Comment 7:

We sincerely thank you for raising this comment regarding our manuscript. It’s a shame that we have not retrieved any literature on the use of pure graphene for functionalized separators in lithium-ion batteries, but the pure GO functional separator [101-103] for lithium ion batteries have been described in the corresponding paragraph.

8.Regarding Comment 8:

We sincerely thank you for raising this comment regarding our manuscript. We have moved Table 4 as required.

9.Regarding Comment 9:

We would like to express our gratitude for highlighting this issue. The diffusion rate of lithium ions through the separator is mainly influenced by the wettability of the separator to electrolyte, now we have added the corresponding description (L676).

10.Regarding Comment 10:

We sincerely appreciate you raising this comment in relation to our manuscript. This paragraph is all about one story of "graphene-based composite interlayer", which is long because of the large number of references cited.

In conclusion, we are grateful for your review comments, and we value your professional insights. We have taken each issue seriously and have made every effort to address them in the revised manuscript. We believe that our revisions have significantly improved the quality and accuracy of the paper.

Reviewer 2 Report

The demand for energy storage has led to mature lithium-ion and metal batteries. However, traditional polymer separators have limitations, hindering battery development. Advanced graphene-based materials offer solutions, improving conductivity, surface area, and mechanical properties. Incorporating them into separators enhances battery capacity, stability, and safety. This review explores graphene's applications in batteries, highlighting advantages and future research directions. The authors did a good job to include most of the related research articles in this review. I think this paper can be published after a minor revision by referring my comments below.

1.     The copyright information should be included in the figure captions since most of the illustrations were reproduced from the published papers.

2.     The author should add the functions of graphene-based materials in the summarized tables for the readers to understand their roles in batteries.

3.     A few important papers should be cited. See below:

(a)   doi.org/10.1038/natrevmats.2016.33

(b)  doi.org/10.3390/c7030065

(c)   doi.org/10.1002/adma.201404210

(d)  doi.org/10.1016/j.joule.2018.07.022

The writing style is okay.

Author Response

Thank you for reviewing our submission and providing valuable feedback and suggestions. We greatly appreciate your expertise, and we will make improvements to the manuscript based on your comments. The corresponding revised manuscript has been uploaded in the attachment.Here is our response to the issues and suggestions you raised:

1.Regarding Comment 1:

We sincerely appreciate you raising this comment. We have carefully reviewed the manuscript and made the necessary revisions. We’ve included the copyright information in figure caption.

2.Regarding Comment 2:

Thank you for bringing up this comment. We’ve included the main functions of graphene-based materials in the summarized tables.

3.Regarding Comment 3:

Thank you for reminding us of this issue. The papers have been cited in the corresponding location as below:

(a)  doi.org/10.1038/natrevmats.2016.33 [90]

(b)  doi.org/10.3390/c7030065 [10]

(c)  doi.org/10.1002/adma.201404210 [145]

(d)  doi.org/10.1016/j.joule.2018.07.022 [159]

In conclusion, we would like to express our gratitude for your review comments, as they are invaluable to us. We have taken each comment seriously and have made all necessary efforts to address them in the revised manuscript. We firmly believe that our revisions have significantly enhanced the quality and accuracy of the paper.
